# Prevalence of Extended-Spectrum β-Lactamases (ESBLs) Producing *Aeromonas* spp. Isolated from *Lamellidens marginalis* (Lamark, 1819) of Sewage-Fed Wetland: A Phenotypic and Genotypic Approach

**DOI:** 10.3390/microorganisms12040723

**Published:** 2024-04-03

**Authors:** Debasmita Mohanty, Basanta Kumar Das, Punam Kumari, Saikat Dey, Asit Kumar Bera, Amiya Kumar Sahoo, Shubhankhi Dasgupta, Shreya Roy

**Affiliations:** 1ICAR-Central Inland Fisheries Research Institute, Barrackpore 700120, West Bengal, India; simidebasmita05@gmail.com (D.M.); asitmed2000@yahoo.com (A.K.B.); amiya7@gmail.com (A.K.S.); shubhankhi56@gmail.com (S.D.); shreyatells@gmail.com (S.R.); 2Department of Bioscience and Biotechnology, Fakir Mohan University, Balasore 756020, Odisha, India; punam.lifescience@gmail.com; 3National Institute of Mental Health and Neurosciences, Bangalore 5600029, Karnataka, India; saikatgene@gmail.com

**Keywords:** zoonotic, *Aeromonas* spp., *L. marginalis*, AMR, extended-spectrum β-lactamase

## Abstract

The global rise of zoonotic bacteria resistant to multiple antimicrobial classes and the growing occurrence of infections caused by *Aeromonas* spp. resistant to β-lactam antibiotics pose a severe threat to animal and human health. However, the contribution of natural environments, particularly aquatic ecosystems, as ideal settings for the development and spread of antimicrobial resistance (AMR) is a key concern. Investigating the phenotypic antibiotic resistance and detection of β-lactamase producing *Aeromonas* spp. in *Lamellidens marginalis*, which inhabit all freshwater ecosystems of the Indian subcontinent, is essential for implications in monitoring food safety and drug resistance. In the present investigation, 92 isolates of *Aeromonas* spp. were recovered from 105 bivalves and screened for their antimicrobial resistance patterns. In vitro antibiotic resistance profiling showed a higher Multiple Antibiotic Resistance (MAR) index of 0.8 with the highest resistance against ampicillin/sulbactam (82%), while 58, 44, 39 and 38% of the isolates were resistant to cephalothin, erythromycin, cefoxitin and imipenem, respectively. PCR results revealed that these isolates carried the bla_TEM_ gene (94%), which was followed by the bla_CTX-M_ gene (51%) and the bla_SHV_ gene (45%). A combination of bla_SHV_, bla_CTX-M_, and bla_TEM_ genes was found in 17% of the isolates, indicating the presence of all three resistance genes. This is the first investigation which highlights the importance of multidrug-resistant *Aeromonas* spp. in *L. marginalis*. The identification of extended-spectrum-β-lactamases (ESBLs) genes demand the necessity of continuous surveillance and systematic monitoring, considering its potential health risks for both animals and human beings.

## 1. Introduction

Antimicrobials have undeniably transformed the field of modern medicine. Nevertheless, their careless use has accelerated the swift emergence of a natural phenomenon: antimicrobial resistance (AMR) in both human and animal pathogens [1]. In response to the growing global human population, the utilization of antimicrobials continues to escalate. From 2000 to 2015, there was a substantial increase (65%) in worldwide antimicrobial consumption, reaching a total of 34.8 billion defined daily doses [2]. As much as 90% of an antimicrobial dose can exit the body unaltered through feces and urine [3]. Subsequently, these excreted antimicrobials have the potential to interact with non-target microorganisms. Furthermore, the release of multidrug-resistant strains from human and animal feces into the environment poses a notable public health hazard [4]. Approximately 1.7 billion individuals lack access to fundamental sanitation, and more than 10% of the global population relies on irrigating crops with wastewater, as reported by the World Health Organization (WHO) in 2022. This agricultural practice also facilitates the transmission of antimicrobial resistance (AMR) among interconnected ecosystems within various natural settings. Different classes of antibiotics are employed to combat microbial infections with β-lactam antimicrobial agents being the most commonly utilized for such purposes. However, the widespread use of β-lactam antimicrobials is currently recognized as the primary cause of resistance to these agents, particularly among Gram-negative bacteria on a global scale [5,6]. *Aeromonas* spp. has the capability to produce multiple β-lactamase enzymes, rendering them resistant to a wide range of β-lactam antimicrobials. This resistance has been increasingly documented in both environmental and clinical isolates [7].

*Aeromonas* spp. exist in a wide host range and hold considerable importance in the context of AMR and noticeable resistance against polymixin B—a highest priority critically important antimicrobials for human medicine (World Health Organization, 2017). Most of the bacteria species under the genus *Aeromonas* exist in the aquatic system without causing any harm. Some of the *Aeromonas* spp. are considered as opportunistic pathogens and cause diseases in both aquatic and terrestrial animals when environmental and host conditions are favorable. They are known to produce extended-spectrum β-lactamases (ESBLs), which is a type of β-lactamase capable of hydrolyzing first, second, and third-generation cephalosporins, and aztreonam. The genes encoding ESBLs can be categorized into various families, including bla_TEM_, bla_SHV_, and bla_CTX-M_ [8]. However, they do not possess the ability to degrade cephamycins or carbapenems. This bacterium has become a promising candidate due to its ability to grow in the host, surrounding environment and artificial media in laboratory conditions. Certain members of this genus have the capacity to infect a wide spectrum of hosts, spanning from cold-blooded to warm-blooded animals, encompassing humans as well [2]. However, given our dependence on both wild and aquaculture food resources, *Aeromonas* spp. pose significant threats to the global economy and food reserves [9]. The aquatic habitat of *Aeromonas* spp. also presents opportunities for human exposure and subsequent infections. Gastroenteritis, septicemia, necrotizing fasciitis, and myonecrosis are among the human diseases caused by *Aeromonas* spp. that present major obstacles to healthcare systems [10]. Given its prevalence in aquatic and human-influenced environments, ease of laboratory cultivation, resistance to crucial antimicrobials, capacity for genetic exchange both within and between species, and ability to cause disease across various animal species, *Aeromonas* stands as a robust and crucial indicator bacterium in monitoring the dynamics of antimicrobial resistance transmission within the framework of One Health [10,11]. *Aeromonas* was found to be considerably more prevalent than *E. coli* in wastewater, freshwater ecosystems, and aquaculture settings [2]. This observation provides additional rationale for utilizing *Aeromonas* spp. as an indicator bacteria for monitoring global antimicrobial resistance (AMR) patterns especially in the context of interconnections between sanitation infrastructure, aquaculture operations, and natural aquatic environments [2].

Global aquaculture production in 2020, as per Food and Agriculture Organization (FAO) data, amounted to 1145 million tonnes, including 177 million tonnes for molluscs like bivalves, while India’s annual wild mussel harvest typically reaches approximately 15,000 tonnes [12]. The freshwater mussel *Lamellidens marginalis* (Lamark, 1819) holds economic significance and is distributed across the freshwater environments of Asia, including India [13]. *L. marginalis* exhibits a broad distribution across wetlands and water bodies in West Bengal and various other states of India. This edible species serves as a traditional source of dietary protein for the rural populations residing in eastern India [13]. *L. marginalis* are suspension feeders that actively filter, capture, and concentrate particles from their aquatic environment. These particles may include contaminants and both free-living and particle-bound microorganisms [14]. *L. marginalis* specimens, collected at random from seemingly healthy populations, frequently harbor a diverse collection of bacteria originating from both aquatic and anthropogenic sources. This bacterial community encompasses a range of species from various genera, including *Vibrio*, *Pseudomonas*, *Moraxella*, *Aeromonas*, *Micrococcus*, *Acinetobacter*, and *Bacillus* [15]. Bivalve molluscs have the potential to be contaminated by these bacterial pathogens, posing significant safety risks when consumed raw or undercooked, as it can lead to human illness. Numerous studies examining the occurrence of pathogenic microorganisms in these bivalve molluscs have been conducted across various countries [7,14,15,16,17,18,19,20]. *Aeromonas* spp. members have demonstrated a propensity to readily acquire single or multiple antibiotic resistance. Furthermore, research by Jones et al. has underscored their significant contribution to the spread of antibiotic resistance within aquatic ecosystems [2].

Sewage fed aquatic bodies receive hazardous chemicals from industrial source and pharmaceutical wastes. *L. marginalis*, a component of such wetlands and being highly sensitive to environmental fluctuations and exposed to pollutants, actively sequesters heavy metals found in suspended particles and deposited in sediment, as well as uptakes emerging pharmaceutical contaminants. This makes them valuable bio-samples for detecting resistant *Aeromonas* spp. These resistant strains can potentially be transmitted to humans and animals through the food chain. While there have been numerous studies on antibiotic-resistant *Aeromonas* spp. in fish, to the best of our knowledge, there is a lack of scientific information on both the phenotypic and genetic characteristics of ESBL-producing *Aeromonas* spp. isolated from *L. marginalis*. We aimed to isolate the bacteria from edible bivalves from sewage-fed wetland and identify antibiotic-resistant microbes from different sources of the aquatic ecosystem with great potential for fisheries resources. The primary objective of this study was phenotypic screening of ESBL-producing *Aeromonas* spp. isolated from *L. marginalis* of a sewage-fed wetland origin. Furthermore, the genotypic screening for ESBL genes of the bacteria was also aimed under study.

## 2. Materials and Methods

### 2.1. L. marginalis Sampling

Monthly sampling were performed from January 2023 to June 2023 over a period of six months. Samples were collected from three waterbodies connected to the sewage canals (Figure 1). A total of 105 live bivalve mussels were gathered from the East Kolkata Wetlands (EKW), which is a recognized Ramsar site situated on the southeastern outskirts of Kolkata, India. The samples were carefully placed in sterile polythene bags and promptly transported to the laboratory in a chilled ice chamber. Subsequently, they were processed within a span of 2 h from the time of collection.

### 2.2. Isolation and Identification of Aeromonas spp.

Each *L. marginalis* sample underwent multiple rinses with PBS (phosphate-buffered saline) to remove any external bacteria. Subsequently, the *L. marginalis*, including both flesh and intravalvular liquid, were aseptically dissected and prepared for bacteriological analysis in accordance with the standard protocol using aseptic techniques (ISO 6887-3:2003 [21]). The samples were homogenized using a tissue homogenizer in a sterile 2 mL tube containing sterile PBS solution (Qiagen Tissue Lyser II, New Delhi, India). Subsequently, 1 g of each composite sample was placed in alkaline peptone water and incubated for 24 h at 37 °C. Following this, the enriched cultures were streaked onto ampicillin dextrin agar (ADA-V) plates (Hi-Media, Mumbai, India) and incubated at 37 °C for 24 h. Presumptive aeromonad colonies displaying the characteristic dark yellow color were selected and subjected to biochemical identification, following established procedures as reported by Dahanayake et al. [17]. Various biochemical tests, including the oxidase test, esculin hydrolases, citrate utilization, and o-nitrophenyl-β-D-galactopyranoside (ONPG), along with the assessment of resistance to the vibriostatic agent 0/129, were conducted for the identification of *Aeromonas* spp. Total DNA extraction followed a rapid boiling procedure [22]. In brief, bacteria were collected from 2 mL of overnight culture, resuspended in 100 µL of sterile distilled water, and lysed by heating at 100 °C for 10 min. The resulting DNA supernatant, obtained through centrifugation, was either used immediately or stored at −20 °C. Confirmation was achieved by partially amplifying genus-specific genes using the 16S rRNA *Aeromonas* genus-specific primers (Table 1). The 25 μL of PCR mixture contains 20–50 ng of bacterial DNA mixed with 10X of PCR buffer, 50 mM of MgCl_2_, 10 mM dNTPs, 5 U/mL of GoTaq^®^ DNA Polymerase (Promega, Madison, WI, USA), and 10 pmol of each primer. Then, the reaction underwent amplification with a hot start at 95 °C for 5 min, which is followed by 35 cycles of denaturation at 95 °C for 30 s, annealing at 58 °C for 1 min, extension at 72 °C for 45 s, and a final extension at 72 °C for 5 min.

### 2.3. Antibiotic Susceptibility Testing

The identified *Aeromonas* spp. strains were cultured in Tryptic Soy Agar (TSA) plates (Hi-Media, Mumbai, India) at 37 °C for 24 h, and the resulting bacterial colonies were transferred to test tubes containing saline solution. The tubes were adjusted visually until they reached a turbidity equivalent to McFarland 0.5. Then, the suspensions were evenly spread onto Mueller–Hinton agar (MHA) plates (Hi-Media, Mumbai, India) using a sterilized cotton swab and incubated at 37 °C for 24 h. The disc diffusion test was conducted and the resistance pattern of the isolates was evaluated following the procedure outlined in the Clinical and Laboratory Standards Institute (CLSI: M39-A4 [26]). A total of twenty antimicrobials comprising eleven distinct antimicrobial classes were employed in the study. These antimicrobials included aminoglycosides (gentamicin 10 μg and amikacin 30 μg), carbapenems (meropenem 10 μg and imipenem 10 μg), β-lactam (cefotaxime 30 μg, cephalothin 30 μg, ceftriaxone 30 μg, cefoxitin 30 μg, ceftazidime 30 μg, and cefepime 30 μg), sulfonamides (trimethoprim–sulfamethoxazole 25 μg), macrolides (erythromycin 15 μg), phenicols (chloramphenicol 30 μg), quinolones (nalidixic acid 30 μg and ciprofloxacin 5 μg), tetracyclines (oxytetracycline 30 μg, tetracycline 30 μg, and doxycycline 30 μg), and β-lactamase inhibitors (amoxicillin/clavulanate 20/10 μg and ampicillin/sulbactam 10/10 μg).

### 2.4. Determination of Multiple Antibiotic Resistance Index (MARI) and Resistance Score (R-Score)

The MARI was determined as per El-Hossary et al. [27] using the formula MARI = a/b, where ‘a’ denotes the count of antibiotics to which the tested *Aeromonas* isolate exhibited resistance, and ‘b’ denotes the total number of antibiotics included in the assessment for the tested *Aeromonas* isolate. In the case of a specific *Aeromonas* isolate, the RI was defined as a quantification representing the count of antibiotics to which the isolate displayed either intermediate or full resistance. The screening system assigned the value of 0.5 and 1 to the isolates that exhibited intermediate and complete resistance, respectively, against a particular antibiotic.

### 2.5. Phenotypic Screening and Confirmation of ESBL Production

For the preliminary screening of ESBL-producing isolates, a method of streaking on HiChrome ESBL agar (Hi-Media, Mumbai, India) was employed, following the procedure outlined by Lamba and Ahammad (2017) [28]. Isolates displaying characteristic blue or bluish-purple colonies were categorized as presumptive ESBL-positive. For further confirmation, the Combination Disc Test (CDT) was employed. In the CDT, discs containing cefotaxime (30 μg) and ceftazidime (30 μg), both with and without clavulanic acid (10 μg), were used. All the test isolates were inoculated onto the MHA (Hi-Media, Mumbai, India) plate, and the antibiotic discs, both with and without clavulanic acid, were put at a distance of 20 mm from each other. Subsequently, the plates were incubated at 37 °C for a period of 48 h, and the resulting zones of inhibition were measured. Isolates were classified as ESBL-positive if the difference in zone diameters between the β-lactam antibiotic disc and the disc containing the antibiotic with clavulanic acid exceeded 5 mm [29].

### 2.6. Detection ESBL Genes (bla_SHV_/bla_CTX-M_/bla_TEM_) by PCR

For the detection of ESBL genes in the phenotypically positive *Aeromonas* spp., all the isolates were screened using three sets of specific primers encoding the gene families bla_SHV_, bla_TEM_ and bla_CTX-M_. The details of theses primers are mentioned in Table 1. Briefly, 2 μL of individual DNA samples, each containing 25–30 ng, was subjected to amplification in a 25 μL PCR reaction mixture. The reaction mixture comprised 10X of PCR buffer, 50 mM of MgCl2, 10 mM of dNTPs, 5 U/mL of GoTaq^®^ DNA Polymerase (Promega, Madison, WI, USA), and both primers at a concentration of 10 nM. The total reaction volume was adjusted by adding 16.8 μL of PCR water. Electrophoresis was performed using a 1.2% agarose gel (Hi-Media, Mumbai, India) in 1X TAE Buffer (Himedia, Mumbai, India). PCR amplicons were visualized in a Gel Documentation System with a UV transilluminator (GelDoc imager, Bio-Rad, Hercules, CA, USA).

### 2.7. Statistical Analysis

In order to compare the frequencies of antimicrobial resistance patterns among ESBL-producing and non-producing *Aeromonas* spp., Chi-square tests (X2) in IBM SPSS version 22.0 was performed. We considered a *p*-value below 0.05 within a 95% confidence interval for establishing statistical significance.

## 3. Results

### 3.1. Identification and Confirmation of Aeromonas spp.

Out of the 105 *L. marginalis* samples, 92 exhibited colonies with a presumptive positive dark yellow appearance on ADA-V plates, which supported the guidelines of the US Environmental Protection Agency from 2001 (Figure 2). Biochemical assessments revealed positive results for oxidase activity, esculin hydrolases, citrate utilization, ONPG, and resistance to vibriostatic agent 0/129. To confirm these findings, further validation was conducted through molecular amplification of the target genes, resulting in an amplicon of 953 bp.

### 3.2. Susceptibility of the Aeromonas spp. to Different Antibiotics

All 92 isolates were screened for resistance against nine antibiotic classes, comprising 20 distinct antibiotics. Remarkably, with the exception of two isolates, all others demonstrated resistance to at least one antibiotic, and notably, four isolates exhibited resistance to more than 11 antibiotics. The highest resistance rate was observed with ampicillin/sulbactam, while the lowest resistance pattern among all isolated *Aeromonas* spp. was noted for ciprofloxacin. Apart from these two antibiotics, other antibiotics- cephalothin, erythromycin, cefoxitin, imipenem, nalidixic acid, gentamycin, amoxicillin/clavulanate, tetracycline, oxytetracycline and doxycycline displayed resistance level varies 58 to 19%. However meropenem, trimethoprim–sulfamethoxazole, ceftazidime and chloramphenicol, cefepime, amikacin, cefotaxime, and ceftriaxone displayed resistance level varies 15 to 8%. These results are given in Table 2.

### 3.3. Phenotypic Screening and Confirmation of ESBLs Producing Aeromonas spp.

In the preliminary screening for ESBL-producing *Aeromonas* spp. on HiChrome ESBL agar (Hi-Media, Mumbai, India), it was observed that among the 92 isolates examined, 55 displayed characteristic purple colonies (Figure 3). These isolates also exhibited ceftazidime (30 μg) inhibition zones of at least 22 mm or cefotaxime (30 μg) inhibition zones of at least 27 mm by the disk diffusion method. Subsequently, these selected isolates were further examined through a phenotypic confirmatory test for ESBL (Figure 3). Utilizing the CDT method on the 55 screened *Aeromonas* spp., positive ESBL activity was identified in 51 isolates, accounting for 55% of the total isolates.

### 3.4. The Antimicrobial Resistance Pattern and the MARI, R-Score of the Isolated ESBL-Producing and Non-Producing Aeromonas spp. to the Tested Antibiotics

In the comparison between ESBL producer and non-producer isolates, the significance levels vary for different antibiotics. These results highlight varying degrees of significance in antibiotic resistance patterns between ESBL-producing and non-producing isolates (Figure 4). Cephalosporins (CEP and CX) exhibited the most significant difference (*p* < 0.001), followed by other antibiotics (CAZ, CTX, CPM, GEN, COT, and E) with a slightly lower level of significance (*p* < 0.01), and finally, ciprofloxacin (CIP) and amikacin (AK) with a somewhat less significant difference (*p* < 0.05). In order to facilitate additional analyses and comparisons, we established both MAR index and R-score values for each of the antibiotic-resistant bacterial isolates. Notably, the highest levels of MAR index and R-score were found in the ESBL isolates, indicating a more pronounced resistance profile. The broader sections of the red violin, corresponding to ESBL negative isolates, signify a higher probability density of populations having a lower MAR index and R-score index. However, ESBL-positive *Aeromonas* spp. exhibit a higher MAR index and R index. These findings are represented in Figure 5, which highlights a highly significant difference with a *p*-value of <0.001 between the ESBL-producing and non-producing isolates. This statistical significance underscores the substantial contrast in antibiotic resistance patterns between the two groups with ESBL-producing isolates displaying a considerably higher degree of resistance.

### 3.5. Resistance Pattern of ESBL-Producing Aeromonas spp. to Different Antibiotic Classes

The study found that a majority of ESBL-producing isolates (70%) were multidrug resistant to three or more antibiotic classes. Specifically, 13 isolates exhibited resistance to two different antibiotic classes, while nine isolates each displayed resistance to three and four different antibiotic classes. Additionally, five isolates were resistant to five different antibiotic classes, seven isolates were resistant to six different antibiotic classes, and three isolates demonstrated resistance to seven antibiotic groups. Notably, there were four isolates that exhibited resistance to eight different antibiotic classes (Table 3).

### 3.6. ESBL Genes and Their Co-Occurrence Harbored by Phenotypically Confirmed ESBL Aeromonas spp.

All the phenotypically confirmed ESBL isolates were genotypically characterized through the PCR method to detect resistance genes. All the 51 phenotypically confirmed ESBL bacteria tested were found positive for at least one ESBL gene, which included bla_SHV_, bla_CTX-M_, or bla_TEM_. Notably, the majority of these isolates carried the bla_TEM_ gene (94%, n = 48), which was followed by the bla_CTX-M_ gene (51%, n = 26), and the bla_SHV_ gene (45%, n = 23). It is observed that 17% of the isolates carried a combination of bla_SHV_, bla_CTX-M_, and bla_TEM_ genes, indicating the presence of all these three resistance genes. In contrast, 4% of the isolates exhibited the co-occurrence of bla_SHV_ and bla_TEM_ gene, suggesting a dual presence of these specific ESBL genes. In addition, 8% of the isolates displayed a combination of bla_TEM_ and bla_CTX-M_ genes, signifying the concurrent presence of these two resistance determinants. Finally, 3% of the isolates demonstrated the simultaneous presence of these particular ESBL genes bla_SHV_ and bla_CTX-M_ (Figure 6).

### 3.7. Combination of ESBL Genes in Association with Resistance to Carbapenem and Non-β-Lactam Antibiotics

Among the 51 ESBL-positive isolates, a significant pattern emerged in terms of the co-occurrence of ESBL genes across different antibiotic classes (Table 4). In the macrolides category, 11 isolates (38%) were found to harbor all three ESBL genes simultaneously. Similarly, within the tetracyclines group, nine isolates (69%) exhibited this co-occurrence pattern. Moving on to the aminoglycosides category, eight isolates (35%) exhibited the presence of all three ESBL genes. The carbapenem and quinolones categories each had seven isolates (35%) that shared this co-occurrence of ESBL genes. Within the sulfonamides and phenicols categories, there were no instances of co-occurrence between bla_SHV_/bla_CTX-M_ and bla_TEM_/bla_CTX-M_. The co-occurrence of bla_TEM_/bla_CTX-M_ was identified in one isolate from each of the carbapenem, tetracycline, and quinolones categories and in two isolates (9%) from aminoglycosides and three isolates (10%) from quinolones. Additionally, one isolate (2%) from the tetracycline and macrolides categories, as well as two isolates from carbapenem and aminoglycosides, demonstrated the co-occurrence of bla_SHV_/bla_CTX-M_. Furthermore, the co-occurrence of bla_TEM_/bla_CTX-M_ was identified in one isolate from each of the carbapenem, tetracycline, and quinolones categories. Moreover, in the aminoglycosides and quinolones categories, two isolates (9%) from aminoglycosides and three isolates (10%) from quinolones demonstrated this co-occurrence pattern.

## 4. Discussion

*Aeromonas* spp. as natural contaminants in bivalves and their surroundings [30]. Contamination with these bacterial species poses a risk to human health, especially through the consumption of aquaculture-derived food [24]. Consuming raw or undercooked bivalves increases the danger of contamination, especially when problems with packaging, transportation, and storage arise, which might result in bacterial infections in humans [31]. Few research studies have been conducted in marine water bivalves, highlighting the significance of detecting *Aeromonas* spp. and their antimicrobial resistance pattern in bivalve molluscs such as clams, mussels, and oysters [17,24,32,33,34,35]. In this study, we emphasized the patterns of antimicrobial resistance in *Aeromonas* spp. recovered from *L. maginalis*, which leads to a significant concern of foodborne *Aeromonas*-related outbreaks. Antibiotic-resistant *A. hydrophila* in vegetables through contaminated water has been linked to significant outbreaks with substantial health risk in China [36]. A growing public health concern centers around the rising prevalence of multidrug-resistant (MDR) *Aeromonas* spp. in sewage water [37,38,39]. In our investigation, *Aeromonas* spp. from freshwater mussels in sewage-fed wetland were targeted and analyzed for the molecular diversity of ESBL genes.

The global incidence of ESBL-producing bacteria is on a rising trend ranging from 33 to 91% in reported cases [40,41,42,43]. Our study identified the presence of ESBL-producing MDR *Aeromonas* strains in the aquatic environment with fisheries operations posing a substantial public health risk. Recent reports of ESBL-producing *Aeromonas* spp. found in food-producing animals and food products also highlighted this concern [4]. To date, there has been limited research on the occurrence of ESBL in *Aeromonas* spp. isolated from bivalves in India. Therefore, the purpose of this study aimed to determine the presence of ESBL in *Aeromonas* spp. from freshwater mussels, utilizing both phenotypic assays and molecular techniques. A significant proportion (55%) of *Aeromonas* isolates from bivalves exhibited positive synergism with the CDT test, indicating the presence of ESBL activity. A study from Taiwan [44] showed that 28% of the clinical isolates were resistant to an extended-spectrum cephalosporin. Our analysis suggests that bivalve mussels have the potential to act as reservoirs for multidrug-resistant bacteria, which is consistent with the findings of Sola et al. [45]. This suggests that increased organic and inorganic contamination in sewage effluents may result in a high load of *Aeromonas* spp. Furthermore, this may play a significant role in promoting the emergence of ESBL-producing *Aeromonas* spp. The presence of such bacteria in bivalve mussels and possibility of transmitting to fish and other aquatic animals cannot be ignored. Similar findings have also been reported in studies conducted in Tanzania and Vietnam [46,47].

Other than the cephalosporins group of antibiotics, we recorded reduced sensitivity to carbapenem antimicrobials, which are known for their effectiveness in treating severe infections caused by ESBL-secreting bacteria [48]. Carbapenem antibiotic is the option for treating bacteria resistant to cephalosporins. In our study, we found that ESBL-producing *Aeromonas* spp. were resistant to both classes of antibiotics, indicating a great concern in treating diseases caused by these resistant strains [49]. This finding is consistent with the results reported by Pfeifer, Cullik, and Witte [50]. Contrary to our results, a study undertaken in China [51] revealed that the majority of isolates (>90%) from clinical, tap water, and food sources were susceptible to carbapenem antibiotics. Our study demonstrated reduced sensitivity to quinolone antibiotics, notably nalidixic acid, following carbapenem antibiotic resistance. Prior research on bivalve molluscs has similarly identified low levels of decreased susceptibility to quinolones, which is a commonly used drug to treat Aeromonas infections in both humans and aquatic animals [16,18]. Contradictory to our result, Deng et al. [52] found that *Aeromonas* spp. isolated from freshwater animals showed more sensitivity toward fluoroquinolones. A small number of isolates (8%) exhibited resistance to chloramphenicol, which is an antibiotic prohibited in food-producing animals. Earlier studies have reported instances of chloramphenicol resistance in bivalve molluscs and fish [15,17,51,53]. The study also found an 18% resistance rate to tetracycline antibiotics, which are widely used in aquatic animal disease control. A study conducted in Turkey on *Aeromonas* spp. isolated from fish samples obtained from local fish markets [53] revealed that all isolates exhibited susceptibility to gentamicin, while a significant proportion (63% to 100%) displayed resistance to trimethoprim–sulfamethoxazole, suggesting that geographic locations and local selective pressure influence the antibiotic resistance levels. In contrast, our investigation found that 27% of the isolates demonstrated resistance to gentamicin, and a minor proportion (13%) exhibited resistance to trimethoprim–sulfamethoxazole. The present investigation was conducted from an aquatic body having a link with city sewage containing various pollutants, including pharmaceutical disposal from clinical settings, which might be a reason for such contrasting results with other reports.

It was found that there is a high occurrence of multidrug resistance among *Aeromonas* spp. isolates with an MAR index of 0.8. This finding indicates that these bacteria have an extensive profile of antimicrobial resistance, as they are resistant to a total of 13 different antibiotics. This study established a significant association between the presence of ESBL-producing *Aeromonas* spp. and both the MAR index and R-score, speculating the significant clinical concern for these resistance patterns. These findings align with the results of a study conducted by Jalil et al. [54], where similar trends of higher MAR index and R-score in *E. coli* isolates were reported. Our finding is consistent with other studies on bacteria of marine bivalves conducted in different parts of Korea [17,32,35,55] with a higher MAR index in *Aeromonas* spp. A study carried out in Malaysia [56] focused on *Aeromonas* spp. isolated from fish samples, revealing an MAR index ranging from 0.07 to 0.64, where 60% of the isolates displayed an MAR index of 0.2 or more. Likewise, we have also recorded MAR index values ranging from 0.05 to 0.8 with 50% of the isolates exhibiting an MAR index exceeding 0.2, which supports the findings of Yi et al. [57] who suggested a positive correlationship with higher MAR index and polluted source of bacteria.

Horizontal gene transfer by bacteria can increase the occurrence of multidrug resistance in aquatic environments [1]. For a better understanding of ESBL resistance, along with phenotypic screening, it is important to detect the responsible resistance genes that may be transferred to other environmental bacteria. ESBLs belong to the class A β-lactamase group, and bla_TEM_, bla_SHV_, and bla_CTX-M_ are the most prevalent ESBL genes found in *Aeromonas* spp. and other Gram-negative bacteria [11,58]. We have detected these three genes with higher prevalence in our isolates. In the present investigation, we observed a predominant presence of *Aeromonas* spp. carrying the bla_TEM_ gene in comparison to the bla_SHV_ or bla_CTX-M_ genes. Previous studies that were carried out on *Aeromonas* spp. obtained from the clinical isolates, tap water, food and cockles of the Korean markets did not find the bla_TEM_ gene [55]. However, in other studies on *Aeromonas* spp. from selfish like manila clam [17], pacific abalone [7] and fish [58] of Korea and Malaysia [56], bla_TEM_ was reported as the most prevalent ESBL gene, which aligns with our findings. The TEM enzymes have a well-established history as a group of β-lactamase associated with ESBL production in bacteria from bivalve samples [15]. Notably, our research identified the second most prevalent ESBL-related gene as bla_CTX-M_ with 51% of ESBL-producing *Aeromonas* spp. after bla_TEM_ (94%). The remarkable mobility of plasmid and transposon-located bla_CTX-M_ genes within successful *E. coli* clones that act as pathogens has established CTX-M β-lactamase as the most dominant ESBL group currently [59]. This finding aligns with other studies that have indicated a high prevalence of *E. coli* producing CTX-M β-lactamase in seafood, which may be influenced by local antibiotic consumption patterns and the circulation of specific resistant clones within the community. The CTX-M group of enzymes, which are commonly found in *E. coli*, confer resistance to cefotaxime. Most of the bla_CTX-M_ isolates were reported to be resistant to cefoxitin due to the loss of a porin in *K. pneumonia* and *E. coli* [59]. However, contradictory to this finding, bla_CTX-M_ carrying *E. coli* isolated from marine bivalves were susceptible to cefoxitin in another study [60]. In our research, we also found the existence of all three ESBL genes together in 31% of the isolates, aligning with the results reported by Hossain and Heo [35]. This raises the possibility that several ESBL genes can also co-occur in a considerable number of *Aeromonas* isolates by pointing to a consistent pattern of ESBL gene distribution in this specific environment.

## 5. Conclusions

Our study highlights the presence of ESBL-producing *Aeromonas* spp. within bivalve molluscs and their potential role in disseminating antibiotic resistance. The findings on antimicrobial resistance in edible bivalve-associated *Aeromonas* spp. further indicates food safety and public health hazards. The present investigation on the antimicrobial resistance of microbes from freshwater bivalve origin results in significant findings specific to a particular geographic area. Hence, we believe that the patterns of resistance may differ between geographic regions and origin of the microbes. To have proper understanding, we propose expanding the investigation covering different animal species and types of microbes. The observed resistance patterns within *Aeromonas* spp. provided evidence toward the contamination of *L. marginalis* from anthropogenic sources. This emphasizes the urgent need for local regulatory authorities to address concerns related to the discharge of untreated or inadequately treated sewage into aquaculture systems to ensure the safety of aquatic environments and public health.

## Figures and Tables

**Figure 1 microorganisms-12-00723-f001:**
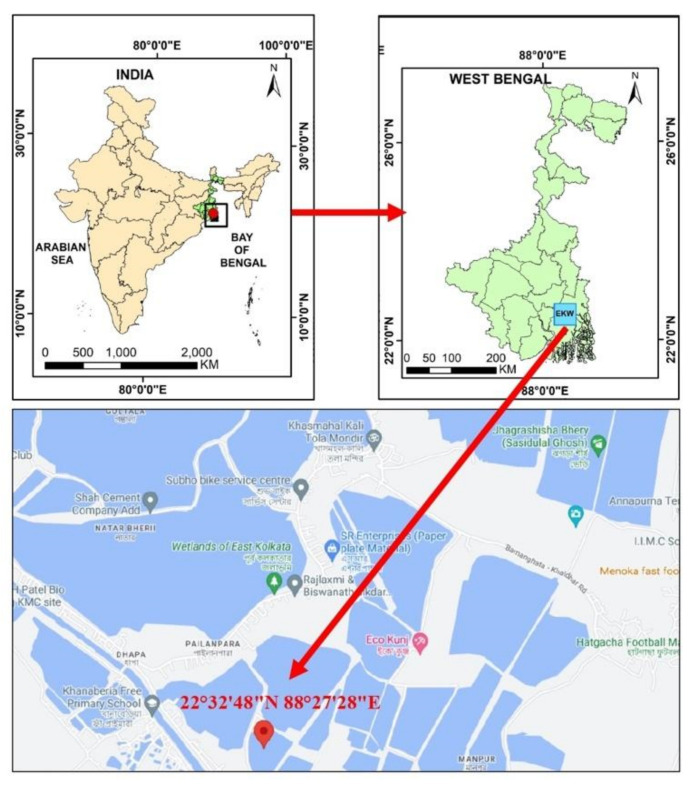
Study area map representing sampling site in EKW, West Bengal, India.

**Figure 2 microorganisms-12-00723-f002:**
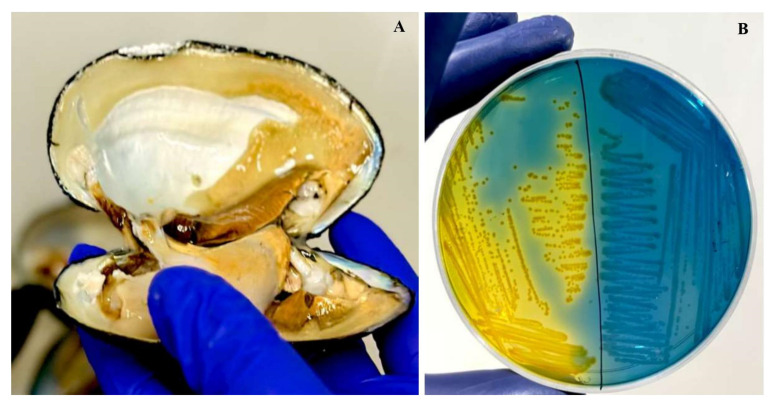
(**A**) Sample collection from *L. marginalis*. (**B**) Colonies of *Aeromonas* spp. with a yellow color colony on ampicillin–dextrin agar base containing ampicillin and vancomycin supplementation.

**Figure 3 microorganisms-12-00723-f003:**
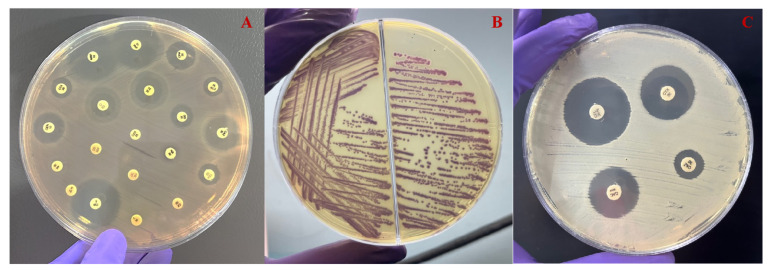
(**A**) The antimicrobial sensitivity test of representative *Aeromonas* spp. reveals resistance to the majority of antimicrobial agents. (**B**) The agar plate depicting the growth of ESBL-producing isolate after 24 h of incubation at 37 °C. (**C**) The CDT method for confirming the production of ESBL production in *Aeromonas* spp. after 48 h of incubation at 37 °C.

**Figure 4 microorganisms-12-00723-f004:**
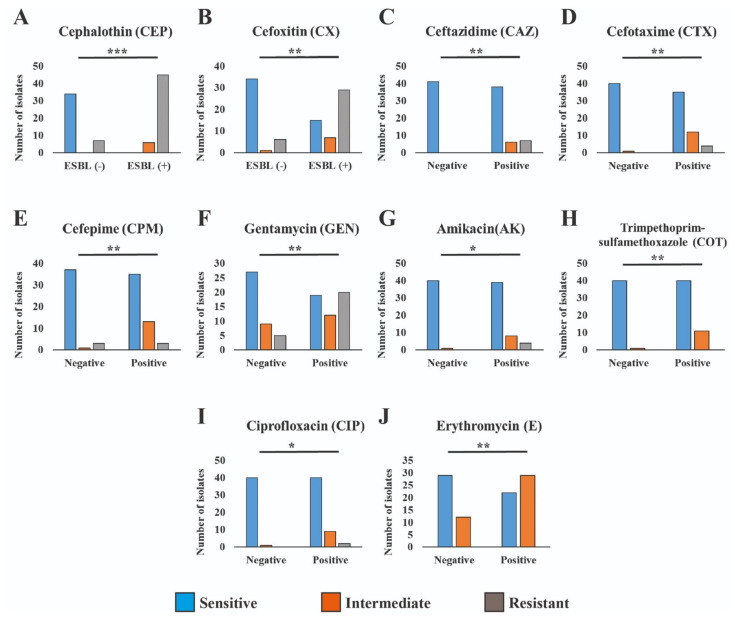
Antimicrobial resistance patterns across ESBL-producing and non-producing isolates differ significantly to varying degrees. The *** refers to *p* < 0.001, ** refers to *p* < 0.01 and * refers to *p* < 0.05.

**Figure 5 microorganisms-12-00723-f005:**
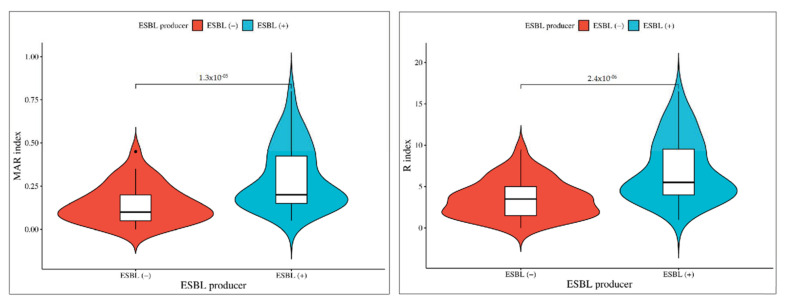
Phenotypic association of isolates between ESBL-producing and non-producing *Aeromonas* spp. The broader sections of the red violin, corresponding to ESBL negative isolates, signify a higher probability density of populations having a lower MAR index and R-score index. However, ESBL-positive *Aeromonas* spp. exhibit a higher MAR index and R index.

**Figure 6 microorganisms-12-00723-f006:**
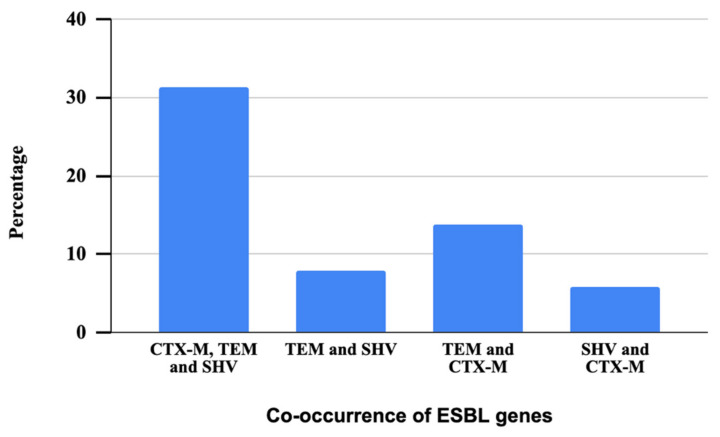
Patterns of co-occurrence of three ESBL genes in ESBL-producing *Aeromonas* spp.

**Table 1 microorganisms-12-00723-t001:** Details of all the primers used in this study to amplify the target genes.

Primer	Fragment	Sequence	Tm (°C)	Products Size (bp)	Reference
16s rRNA	Forward	5′-AAGAGTTTGATCCTGGCTCAG-3′	56	953	[23]
	Reverse	5′-GGTTACCTTGTTACGACTT-3′			
TEM	Forward	5′-CATTTCCGTGTCGCCCTTATTC-3′	64	800	[17]
	Reverse	5′-CGTTCATCCATAGTTGCCTGAC-3′			
SHV	Forward	5′-AGCCGCTTGAGCAAATTAAAC-3′	52	713	[24]
	Reverse	5′-ATCCCGCAGATAAATCACCAC-3′			
CTX-M	Forward	5′-CGCTTTGCGATGTGCAG-3′	50	550	[25]
	Reverse	5′-ACCGCGATATCGTTGGT-3′			

**Table 2 microorganisms-12-00723-t002:** The in vitro antibiotic susceptibility pattern of 92 isolates, illustrating the percentage of resistance and sensitivity of *Aeromonas* spp. to the tested antibiotics.

Antibiotics	R	I	S
CEP	58	7	39
CX	39	9	55
CTR	3	4	96
CAZ	8	7	89
CTX	4	14	85
CPM	7	15	82
IPM	38	28	37
MRP	15	7	82
AMC	22	28	53
A/S	82	10	12
GEN	27	23	53
AK	4	10	89
TE	20	0	84
O	18	0	85
DO	18	0	85
COT	13	0	90
CIP	2	11	90
C	8	0	96
E	45	0	59
NA	35	2	66

Abbreviations: cephalothin (CEP), cefoxitin (CX), ceftriaxone (CTR), ceftazidime (CAZ), cefotaxime (CTX), cefepime (CPM), imipenem (IMP), meropenem (MRP), amoxicillin/clavulanate (AMC), ampicillin/sulbactam (A/S), gentamicin (GEN), amikacin (AK), tetracycline (TE), oxytetracycline (O) doxycycline (DO), trimethoprim–sulfamethoxazole (COT), ciprofloxacin (CIP), chloramphenicol (C), erythromycin (E) nalidixic acid (NA)

**Table 3 microorganisms-12-00723-t003:** The antibiogram pattern of ESBL-producing *Aeromonas* spp. exhibiting resistance to various antibiotics and presence of ESBL genes.

Number of Isolates	Resistance Phenotypes	Intermediate	No. of Drug Class	ESBL Gene Determinants
1	CEP	-	1	TEM-CTX-M
2	CEP-CAZ-A/S	-	2	SHV-CTX-M
1	CEP-CX-A/S	-	2	TEM-CTX-M
1	CEP-CX-A/S	CPM-AMC-GEN	2	TEM
1	CEP-CX-E	GEN	2	TEM-CTX-M
1	CEP-A/S	IPM	2	SHV-TEM-CTX-M
1	CEP-CX-E	CTX-GEN-AK	2	SHV-TEM-CTX-M
2	CEP-A/S-E	CX-GEN	3	TEM
1	CEP-CX-AMC	CPM-A/S	2	TEM
1	CEP-A/S	CTX-AMC	2	TEM
1	CEP-CX-AMC-A/S	-	4	TEM-CTX-M
1	CEP-NA-E	CX-IPM-AMC-A/S	3	TEM
1	CEP-CX-AMC-NA	IPM-A/S	3	SHV-TEM-CTX-M
1	CEP-A/S-GEN-E	CTX-IPM	4	SHV-TEM-CTX-M
1	CEP-CX-A/S-GEN	AK	4	TEM
1	IPM-A/S	CEP-MRP-AMC	2	TEM
1	A/S-GEN-NA-E	CEP-CX-CAZ-CPM	4	TEM
1	CEP-CX-A/S-NA	CPM-IPM-AMC-GEN	3	TEM
1	CEP-IPM-MRP-AK	CX-CPM-AMC-A/S-GEN	3	TEM
1	IPM-MRP-A/S-GEN	CEP-AMC	3	SHV-TEM-CTX-M
1	A/S-GEN	CEP-IPM	2	TEM-CTX-M
1	CEP-CX-A/S-NA	CTX-IPM	3	SHV-TEM
1	IPM-A/S-GEN	CEP-MRP-AMC	3	TEM
1	CEP-IPM-MRP-A/S-GEN	CPM	4	TEM
1	CEP-CX-A/S-GEN-E	CTX-AK	2	TEM
1	CEP-CX-A/S-NA-E	IPM	4	TEM
1	CEP-CX-IPM-A/S-AK-E	AMC	5	TEM
1	CEP-IPM-A/S-GEN-NA-E	CX-AK-CIP	5	TEM
1	CEP-CTR-IPM-A/S-COT-NA-C	CX	4	SHV-TEM
1	CEP-CX-CAZ-CTX-A/S-GEN-NA-E	CTR	5	TEM
1	CEP-CX-CTX-A/S-CIP-NA-E	CTR-CAZ-IPM-GEN	4	TEM
1	IPM-MRP-A/S-GEN-TE-O-DO-E	CEP-AK	5	SHV-CTX-M
1	CEP-CX-IPM-AMC-A/S-GEN-COT-E	CPM-AK	6	SHV-TEM-CTX-M
1	CEP-CX-CAZ-CTX-AMC-A/S-COT-CIP-C	CTR-CPM-IPM-GEN	5	SHV-TEM-CTX-M
1	CEP-CX-IPM-MRP-A/S-TE-O-DO-NA	AMC-CIP	4	SHV-TEM-CTX-M
1	CEP-CAZ-A/S-TE-O-DO-COT-NA-E	-	6	SHV-TEM-CTX-M
1	CEP-CX-AMC-A/S-TE-O-DO-NA-C-E	GEN	6	TEM
1	CEP-CX-AMC-A/S-TE-O-DO-COT-NA-C-E	IPM-GEN	6	SHV-TEM-CTX-M
1	CEP-CX-A/S-TE-O-DO-COT-NA-E	CTX-CPM-IPM-CIP	6	SHV-TEM
1	CEP-IPM-A/S-GEN-AK-TE-O-DO-COT-E	CPM-AMC-CIP-NA	7	SHV-TEM-CTX-M
1	CEP-IPM-MRP-AMC-A/S-GEN,TE-O-DO-COT-NA-E	CAZ-CTX-CIP	8	TEM-CTX-M
1	CEP-IPM-MRP-AMC-A/S-GEN,TE-O-DO-COT-NA-E	CTX	7	SHV-TEM-CTX-M
1	CEP-CX-CAZ-CPM-IPM-AMC-A/S-GEN-TE-O-DO-E	CTX	6	SHV-TEM-CTX-M
1	CEP-CX-CPM-IPM-AMC-A/S-GEN-TE-O-DO-COT-NA-C-E	CAZ-CIP	8	SHV-TEM-CTX-M
1	CEP-CX-CTR-CAZ-CTX-CPM-AMC-A/S-AK-TE-O-DO-COT-NA-C-E	GEN	8	SHV-TEM-CTX-M

**Table 4 microorganisms-12-00723-t004:** The presence of ESBL genes in combination with resistance to carbapenem and non-β-lactam antibiotics.

No. of Isolates Resistant to Carbapenem Non-β-Lactam Antibiotics	Pattern of ESBL Genes			
	CTX-M, TEM and SHV (%)	SHV and TEM (%)	TEM and CTX-M (%)	SHV and CTX-M (%)
Carbapenems (n = 18)	39	6	6	11
Aminoglycosides (n = 23)	35	0	9	9
Tetracyclines (n = 13)	69	8	8	8
Sulfonamides (n = 11)	36	9	0	0
Quinolones (n = 20)	35	15	5	0
Phenicols (n = 6)	67	17	0	0
Macrolides (n = 29)	38	3	10	3

## Data Availability

Data are contained within the article.

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
