# Peer review of "Prevalence of Extended-Spectrum β-Lactamases (ESBLs) Producing Aeromonas spp. Isolated from Lamellidens marginalis (Lamark, 1819) of Sewage-Fed Wetland: A Phenotypic and Genotypic Approach"

_microorganisms, 2024, doi:10.3390/microorganisms12040723_

Round 1
Reviewer 1 Report
Comments and Suggestions for Authors
The title of the manuscript describes fully the scope of the paper. The topic is sufficiently important and the paper has its merits. However, quite a bit of rearrangements are needed.
Two main points:
- Why so many ESBL-producers are susceptible to B-lactams (Fig. 4) ? By definition these isolates should be resistant. The sensitivity might be an artefact (see the expert opinions at the EUCAST website).
- The results section needs total reorganisation. Now ESBL producers are discussed before the results have been given, this is very confusing, I had to read through the results several times. Also, the figure numbers are wrong.
Other points:
- lines 13-14: “pose a widespread issue” : this sounds very colloquial. Be more specific.
- line 18: “essential for implications in” : as above
- line 19: specify how many isolates
- line 25: presence instead of “identification”
- line 27: “potential health risks for both aquatic animals” : what kind of health risks?
- lines 38 & 41: no need to mention the names of the authors, the ref. number is enough
- line 50 “generate” : produce?
- line 59 “gained recognition as a promising candidate”: for what?
- lines 71-74: important point but now stated twice, merge the sentences.
- Lines 75-76: give a ref.
- Lines 94-95 “these bacterial pathogens” the mentioned genera are not all pathogens, just some subtypes, most of them are normal bacteria found everywhere
- line 154: “β-lactamase inhibitors”: B-lactamases?
- line 162: check the ref (remove initials) and in that paper I don’t really find any method, just plating on Hi-Media. This is important information though, since I am quite sceptical about results obtained using Hi-Media. Even my collaborators in resource-poor settings don’t use Hi-Media, even if it is much cheaper than the similar products from other producers. When comparing the results from media produced by different manufacturers to other methods, Hi-Media seems to often give very unreliable results.
- line 169 “RI” what’s this?
- line 186: using instead of “for”
- line 187, correct the sentence
- line 206: here should be figure 3. In any case, the gel figure should go to supplementary material/appendix.
- line 212: no need to repeat
- line 225 “These results are visualized in figure 4.” I assume you refer to Figure 3 here. Turn the figure into a table, and delete the deleted information from the text.
- line 225-226 “ESBL producer and non-producer isolates” this figure is in a wrong place, since the ESBL-production results are given only later.
- lines 239-240: rewrite the title, use Aeromonas spp to be more specific
- Figure 5: needs to be explained more, I don’t understand it.
- Table 2: add precision to the title, give a number of isolates, since it is very difficult to follow which subset is discussed in different parts of the text.
- lines 285-292: this chapter is very simplistic, shorten and combine into the chapter 3.7
- Discussion: revise, now unnecessary information and repetition
- Many refs indicated in the discussion actually talk about E. coli. Are they relevant to this paper? Consider removing.
- lines 389-391: give a ref to the statement.
- Conclusion: again repetition, eg. the two first sentences
- lines 408-409: “Antimicrobial resistance in these bivalves needs to expand the idea in certain geographic areas and among other animal species.”: I don’t understand the sentence.
- line 410: “provided hits towards”: what do you mean?
- line 413: food safety
- Ref. 44: edit
Comments on the Quality of English LanguageIn general the English is good, but some poor word choices and unclear sentences.
Author Response
|
The authors are highly obliged for your important corrections to improve the manuscript |
||
|
sl no |
Comment |
Response |
|
1 |
Why so many ESBL-producers are susceptible to B-lactams (Fig. 4)? By definition these isolates should be resistant. The sensitivity might be an artefact (see the expert opinions at the EUCAST website). |
We have done the AST and ESBL production of the isolates as per the standard protocol of CLSI guidelines (CLSI, 2009). As described by Wu et al., 2011, Aeromonas isolates that demonstrated a diameter of inhibition zone of ceftazidime (30 μg) of ≤22 mm or of cefotaxime (30 μg) of ≤27 mm by the disk diffusion method—i.e., reduced susceptibility to expanded-spectrum cephalosporins—were examined by a phenotypic confirmatory test for ESBL production. Hence, all ESBL producers may not be beta-lactams resistant. |
|
2 |
The results section needs total reorganisation. Now ESBL producers are discussed before the results have been given, this is very confusing, I had to read through the results several times. Also, the figure numbers are wrong. |
The results section is reorganised as per the suggestion. The corrections are as follows:
3.2. Susceptibility of the Aeromonas spp. to different antibiotics 3.3. Phenotypic screening and confirmation of ESBLs producing Aeromonas spp. 3.4. The antimicrobial resistance pattern and the MARI, R-score of the isolated ESBL producer and non-producer bacteria to the tested antibiotics 3.5. Resistance pattern of ESBL producing Aeromonas spp. to different antibiotic classes 3.6. ESBL genes and their co-occurrence harboured by phenotypically confirmed ESBL Aeromonas spp. 3.7. Combination of ESBL genes in association with resistance to carbapenem and non- β-lactam antibiotics
The ESBL producers mentioned in the 3.2 subsection is now merged in 3.4 after the screening and confirmation of ESBLs producing Aeromonas spp.The figures were also mentioned in the respective subsections. |
|
3 |
lines 13-14: “pose a widespread issue”: this sounds very colloquial. Be more specific. |
The sentence has been modified. |
|
4 |
- line 18: “essential for implications in” : as above |
The sentence has been modified as “ are essential for implications in monitoring food safety and drug resistance.” |
|
5 |
- line 27: “potential health risks for both aquatic animals” : what kind of health risks? |
The potential health risks mentioned in the sentence refer to the threats that can arise as a result of disease caused by bacteria having extended-spectrum-β-lactamases (ESBLs) genes in aquatic animals and also human beings. ESBLs are enzymes that confer resistance to a broad range of antibiotics, particularly β-lactam antibiotics. The detection of these genes suggests that bacteria in the aquatic environment and animals, may have resistance ability against those antibiotics. |
|
6 |
- lines 38 & 41: no need to mention the names of the authors, the ref. number is enough |
Names of the authors are removed. |
|
7 |
- line 50 “generate” : produce? |
The word is replaced by “produce” |
|
8 |
- line 59 “gained recognition as a promising candidate”: for what? |
Aeromonas spp., widely distributed across various environments and serving as both a pathogenic entity and reservoir for antibiotic resistance genes, makes them a good candidate for screening and characterizing antibiotic resistance profiles. |
|
9 |
- lines 71-74: important point but now stated twice, merge the sentences. |
These sentences are merged. |
|
10 |
- Lines 75-76: give a ref. |
Reference is cited. |
|
11 |
- Lines 94-95 “these bacterial pathogens” the mentioned genera are not all pathogens, just some subtypes, most of them are normal bacteria found everywhere |
Yes, we agree with the comment provided by the reviewer. Now we have incorporated the same in line no 62-66 as given below: Most of the bacteria species under the genus Aeromonas exist in the aquatic system without causing any harm. Some of the Aeromonas spp. are considered as opportunistic pathogens and cause diseases in both aquatic and terrestrial animals when environmental and host conditions favours. |
|
12 |
- line 154: “β-lactamase inhibitors”: B-lactamases? |
These antibiotics were come under the β-lactam group. The word is changed to “β-lactam” |
|
13 |
- line 162: check the ref (remove initials) and in that paper I don’t really find any method, just plating on Hi-Media. This is important information though, since I am quite sceptical about results obtained using Hi-Media. Even my collaborators in resource-poor settings don’t use Hi-Media, even if it is much cheaper than the similar products from other producers. When comparing the results from media produced by different manufacturers to other methods, Hi-Media seems to often give very unreliable results. |
The reagents, especially the bacteriological media and antibiotic sensitivity assay of HiMedia are well known for their good results, in most of the reputed laboratories in India. We are getting standard results using the chemicals. We agree that there are better firms with good quality chemicals/reagents, but the results obtained using the HiMedia make chemicals were satisfactory since in each set of experiments we kept the control bacteria isolates (ATCC).
|
|
14 |
- line 169 “RI” what’s this? |
RI refers to “Resistance-score Index” |
|
15 |
- line 186: using instead of “for” |
The suggestion is made in the materials and method section.
|
|
16 |
- line 187, correct the sentence |
Corrected the sentence and include the details of the PCR master mix. |
|
17 |
- line 206: here should be figure 3. In any case, the gel figure should go to supplementary material/appendix. |
All the gel photos are merged together. However as per suggestion those are given in as supplementary file
|
|
18 |
- line 212: no need to repeat |
Deleted the repeated sequence. |
|
19 |
- line 225 “These results are visualized in figure 4.” I assume you refer to Figure 3 here. Turn the figure into a table, and delete the deleted information from the text. |
The antibiotic susceptibility results are described in a table in place of stack chart and mentioned in table 2. |
|
20 |
- line 225-226 “ESBL producer and non-producer isolates” this figure is in a wrong place, since the ESBL-production results are given only later. |
After the rearrangement of the result section, figure 4 placed after the screening of ESBL producing Aeromonas spp. (3.4 subsection). |
|
21 |
- lines 239-240: rewrite the title, use Aeromonas spp to be more specific |
As suggested by the Reviewer the title of the table is modified as “The antimicrobial resistance pattern and the MARI, R-score of the isolated ESBL producing and non-producing Aeromonas spp. to the tested antibiotics” |
|
22 |
- Figure 5: needs to be explained more, I don’t understand it. |
In figure 5, a violin plot represents a mirrored violin-shaped figure, which explains the kernel density estimation (a method for examining and comprehending data distribution shapes) is depicted. The broader sections of the red violin, corresponding to ESBL negative isolates, signify a higher probability density of populations having a lower MAR index and R index. However, ESBL positive Aeromonas spp. exhibit a higher MAR index and R index which highlights a highly significant difference with a p-value of < 0.001. Within the violin, the enclosed box plot summarizes the distribution of the data. This is added in the respective result section |
|
23 |
- Table 2: add precision to the title, give a number of isolates, since it is very difficult to follow which subset is discussed in different parts of the text. |
Table 2 is changed to Table 3 after the modifications of the result section. The title is also modified as suggested. Isolate numbers are added in different column for each resistance type in the modified table. |
|
24 |
- lines 285-292: this chapter is very simplistic, shorten and combine into the chapter 3.7 |
3.6 and 3.7 subsections are merged and named as “ESBL genes and their co-occurrence harboured by phenotypically confirmed ESBL Aeromonas spp.” under 3.6. |
|
25 |
- Discussion: revise, now unnecessary information and repetition |
Discussion sections have been modified as suggested by the reviewer. |
|
26 |
- Many refs indicated in the discussion actually talk about E. coli. Are they relevant to this paper? Consider removing. |
Since, the references are relevant to the antimicrobial resistance pattern involving the ESBL producing E. coli and Klebsiella., and those were exactly not available for Aeromonas spp. However, we have removed some of them except the most relevant references |
|
27 |
- lines 389-391: give a ref to the statement. |
Reference has been added in the text |
|
28 |
- Conclusion: again repetition, eg. the two first sentences |
These sentences are modified as per the suggestion. |
|
29 |
- lines 408-409: “Antimicrobial resistance in these bivalves needs to expand the idea in certain geographic areas and among other animal species.”: I don’t understand the sentence. |
The sentence is modified and incorporated in the respective part of the conclusion as given below: “The present investigation on antimicrobial resistance of microbes from freshwater bivalve origin leads to the important findings for a particular geographical region. Hence, we opine that the patterns of resistance may differ between geographic regions and the origin of the microbes. To have a proper understanding, we propose expanding the investigation covering different animal species and types of microbes.” |
|
30 |
- line 410: “provided hits towards”: what do you mean? |
The word “hits” is written by typographical mistake in place of “hints”. It is corrected in the text. |
|
31 |
- line 413: food safety |
Here we have mentioned the safety of aquatic environment considering drug resistance among the bacteria available within the system as a whole. When resistant bacteria are available in the edible component of aquatic systems that may lead further public health concerns. |
|
32 |
- Ref. 44: edit |
The reference is edited. |

Reviewer 2 Report
Comments and Suggestions for Authors
The authors reported on the prevalence of extended-spectrum β-lactamases (ESBLs) producing Aeromonas spp. isolated from Lamellidens marginalis (Lamark, 1819) of sewage fed wetland using a phenotypic and genotypic approach. No justification why this work could increase our understanding of antibiotic resistance and no working hypothesis have been presented. There is a multitude of reports on the antibiotic resistances of pathogenic bacteria, many of them characterized the responsible genes as well. Here are some recent examples:
Jones, D. C., LaMartina, E. L., Lewis, J. R., Dahl, A. J., Nadig, N., Szabo, A., ... & Skwor, T. A. (2023). One Health and Global Health View of Antimicrobial Susceptibility through the “Eye” of Aeromonas: Systematic Review and Meta-Analysis. International Journal of Antimicrobial Agents, 106848.
Yuan, X., Lv, Z., Zhang, Z., Han, Y., Liu, Z., & Zhang, H. (2023). A Review of Antibiotics, Antibiotic Resistant Bacteria, and Resistance Genes in Aquaculture: Occurrence, Contamination, and Transmission. Toxics, 11(5), 420.
Adah, D. A., Saidu, L., Oniye, S. J., Adah, A. S., Daoudu, O. B., & Ola-Fadunsin, S. D. (2024). Molecular characterization and antibiotics resistance of Aeromonas species isolated from farmed African catfish Clarias gariepinus Burchell, 1822. BMC Veterinary Research, 20(1), 1-14.
Odeyemi, O. A., Amin, M., Dewi, F. R., Kasan, N. A., Onyeaka, H., Stratev, D., & Odeyemi, O. A. (2023). Prevalence of Antibiotic-Resistant Seafood-Borne Pathogens in Retail Seafood Sold in Malaysia: A Systematic Review and Meta-Analysis. Antibiotics, 12(5), 829.
Mollusks have also been assessed for disseminating antibiotic resistances. Concerning these reports, the study does not have any novelty. The study is also lacking a proper placement of the findings in the context of similar reports. Is there any differences in the spectrum of antibiotic resistances of Aeomonas spp. from Lamellidens marginalis to those from other sources, especially to clinical isolates? Is there a difference to other parts of India or Asia and if so, why? Is there a connection between the use of antibiotics in this area and the antibiotic resistances detected in the mollusks? How is the sewage input into the sampling sites? With other words: What can be learned from this study which has not been known before?
Abstract: what is MAR? Typing error of AMR?
Lines 58ff: Please be aware that the singular of bacteria is bacterium and correct the text accordingly.
How many Aeromonas isolates have been obtained, how many of them have been investigated? Are the results really based on only 55 isolates?
Please be realistic when given the percentages of the isolates: if 55 isolates have been investigated no decimals in the percentages can be given because they are not reproducible. Therefore, delete all decimals in these numbers.
The discussion should be a discussion of the results and not another introduction. For this, the discussion section has to be streamlined, e.g. by deleting the first paragraph, which is merely a repeat from the introduction.
Author Response
|
Reviewer 2: We are thankful for your suggestions. Now we have improved the manuscript, considering the concerns raised by you. |
||
|
sl no |
Comment |
Response |
|
1 |
The authors reported on the prevalence of extended-spectrum β-lactamases (ESBLs) producing Aeromonas spp. isolated from Lamellidens marginalis (Lamark, 1819) of sewage fed wetland using a phenotypic and genotypic approach. No justification why this work could increase our understanding of antibiotic resistance and no working hypothesis have been presented. |
In this study, we approached isolating the bacteria from edible bivalves from a sewage fed wetland and pointed out a different source/reservoir of antibiotic resistant microbes in the aquatic system with great potential for fisheries resources. The introduction section is now modified with a proper working hypothesis. The modification done in the introduction as per suggestion |
|
2 |
Abstract: what is MAR? Typing error of AMR? |
MAR stands for Multiple Antibiotic Resistance. This an index for a bacterium isolate showing resistance to more than two antibiotics. If an isolate is resistant to X nos. of antibiotics out of Y Nos antibiotics used, the MAR=X/Y. |
|
3 |
Lines 58ff: Please be aware that the singular of bacteria is bacterium and correct the text accordingly. |
The correction is made in the text as suggested |
|
4 |
How many Aeromonas isolates have been obtained, how many of them have been investigated? Are the results really based on only 55 isolates? |
From 105 bivalve samples, a total of 92 isolates of Aeromonas spp. were recovered. Out of 92 isolates, 55 were ESBL producers and the remaining 37 were identified as non-ESBL producers. All the 55 ESBL producing isolates were further considered for ESBL gene detection and co-occurrence of resistance with other class of antibiotics. |
|
5 |
Please be realistic when given the percentages of the isolates: if 55 isolates have been investigated no decimals in the percentages can be given because they are not reproducible. Therefore, delete all decimals in these numbers. |
The total ESBL producing isolates were 55 Nos. They were classified into different non beta lactam groups and carbapenem group resistant. Under each class, percentages were made. However, during calculation the total were considered 55 in all the cases by mistake instead of ‘n’ value of each respective row. Hence, the table is corrected and replaced by the modified one. The changed values are given in the respective results and discussion section. We agree that the decimal value for isolate can not be reproduced. The values are modified to near most whole numbers as per the suggestion. |
|
6 |
The discussion should be a discussion of the results and not another introduction. For this, the discussion section has to be streamlined, e.g. by deleting the first paragraph, which is merely a repeat from the introduction. |
Discussion sections have been modified as suggested by the reviewer. |

Reviewer 3 Report
Comments and Suggestions for Authors
The present study describes the prevalence, AMR and ESBL features of Aeromonas spp. isolates from Lamellidens marginalis. It is an interesting study however significant improvements are needed to proceed further with the manuscript.
Abstract. Please include the information about samples and isolates used in the present study.
Not all Aeromonas spp. are pathogenic to humans or animals, this needs to be properly addressed in the Introduction.
Line 53. "Aeromonas spp. are classified as critical priority pathogens". I was not able to find Aeromonas spp. in the list of WHO priority pathogens.
Line 58-60. The sentence is not clear. Please rewrite.
Line 60. The authors should link the Aeromonas spp. species with pathologies they cause in humans and animals.
Line 65. Do the mentioned diagnoses require antimicrobial therapy? The authors need to prove that AMR in Aeromonas spp. pose animal or public health issues.
Line 69-70. What are the crucial antimicrobials? What is a genetic exchange? Between which species infections may be caused? This needs further explanation.
Line 68-74. The easiness of incubation is repeated twice in two sentences. Could the authors provide more explanations for the application of Aeromonas spp. as AMR indicators in the environment?
Line 74. How prevalent is Aeromonas spp. in terrestrial ecosystems?
Line 80. FAO. Please provide the full name for an abbreviation.
Line 110. What do the authors aim to detect with "phenotypic screening of ESBL producing Aeromonas spp."?
Materials and Methods.
2.1. Sampling.
Please add the period when samples were collected/the study was conducted. Add no. of samples per location? How many sampling sessions were in total? How many mussels were collected? From water or cost?
2.2. In paper no. 17 biochemical confirmation procedure of Aeromonas spp. isolates is not described.
Very confusing that the authors provide a reference to Table 1 only for 2.2. subsection. PCR mix and reaction conditions are not provided.
2.3. Please provide culturing conditions for inoculated TSA and MHA agars. Please check manufacturer information for all microbiological media.
2.3., 2.5. Please merge phenotypical examinations of ESBL isolates in one subsection.
Results.
3.1. Biochemical procedures were not described in the M and M section.
3.2. Difficult to follow results on antimicrobial resistance. Please show them in a table.
Regarding 92 isolates, how were they obtained? From culturing or they were PCR confirmed? How many isolates were confirmed from one sample?
Line 229. P less than 0.05 please use symbols.
Fig.2.
Fig.3. What is the difference between ESBL(-) and ESBL(+) and negative and positive?
Line 254. I cannot find bluish colonies in Fig.5.
Fig.6. any specific detail on antibiotics, incubation conditions, or isolates is not provided.
Table 2. Explanations for antimicrobials are not provided. What do the authors aim to show with this table? Information on specific ESBL determinants could be included.
Merge 3.6. and 3.7. subsections.
3.8. subsection should be merged with 3.5., 3.6. and 3.7.
Discussion.
Line 323. "these bacterial spp. poses". Please explain.
Line 330-331. Please add information on foodborne outbreaks of Aeromonas spp. caused by resistant Aeromonas spp.
Line 335-336. In mussels? In waters? Not clear how this relates to the present manuscript.
Line 336-337. The present study aimed to detect ESBL Aeromonas spp. in Lamellides but not in the environment.
Line 338-339. Have the authors studied ESBL production in Aeromonas spp. isolates? What kind of concerns? From which foods and food-production animals were resistant Aeromonas spp. isolated? Have the authors draw this conclusion from one study?
Line 351. What kind of findings?
Line 353. Non beta-lactam antibiotics...carbapenems. Please check.
Line 358-360. The sentence is unclear.
Line 397-398. How this was addressed by the present study?
The discussion describes general information about ESBL-producing microorganisms with little relevance to the present study.
Conclusions.
Line 404-408. Two sentences with similar meaning.
Aeromonas spp. are autochthonous microorganisms to aquatic environments. How was it highlighted in the Introduction, Discussion and Conclusions?
Comments on the Quality of English LanguageExtensive English revisions are needed.
Author Response
|
Reviewer 3: We are thankful to the reviewer for pointing out the errors which is really helpful in improving the manuscript |
||
|
sl no |
Comment |
Response |
|
1 |
Abstract. Please include the information about samples and isolates used in the present study. |
The information related to sample size and total no of Aeromonas isolates is added in the abstract part as: In the present investigation 92 isolates of Aeromonas spp. were recovered from a sample of 105 bivalves samples and screened for their antimicrobial resistance patterns. |
|
2 |
Not all Aeromonas spp. are pathogenic to humans or animals, this needs to be properly addressed in the Introduction. |
Incorporated in the introduction section. |
|
3 |
Line 53. "Aeromonas spp. are classified as critical priority pathogens". I was not able to find Aeromonas spp. in the list of WHO priority pathogens. |
The sentence was written considering Aeromonas as pathogen for wide host range and its importance in AMR even its recent trend on resistance against polymixin B-the highest priority critically important antimicrobial for human medicine” (World Health Organization, 2017). However the sentence is now modified accordingly |
|
4 |
Line 58-60. The sentence is not clear. Please rewrite. |
The sentence is rewritten as: This bacterium has become a promising candidate due to its ability to grow in both host, surrounding environment and artificial media in laboratory condition. |
|
5 |
Line 60. The authors should link the Aeromonas spp. species with pathologies they cause in humans and animals. |
The information is provided in the introduction at line 72 and 74. |
|
6 |
Line 65. Do the mentioned diagnoses require antimicrobial therapy? The authors need to prove that AMR in Aeromonas spp. pose animal or public health issues. |
Clinical forms of Aeromonas infections need antimicrobial therapy in humans, animals and fish. Aeromonas infections are reported in both humans and animals. The antimicrobial resistance can be transmitted with in Aeromonas spp and also other pathogenic bacteria through horizontal transfer of AMR genes. Hence, aquatic ecosystems with antimicrobial resistant Aeromonas spp may pose a risk for surrounding aquatic and terrestrial animals and also for public health. However, in this research work sample analysis from animals and humans was beyond the scope. |
|
7 |
Line 69-70. What are the crucial antimicrobials? What is a genetic exchange? Between which species infections may be caused? This needs further explanation. |
Aeromonas isolates exhibit resistance to many broad-spectrum antibiotics, including beta-lactam, beta-lactamase inhibitor, carbapenem, tetracycline, and quinolones on which we rely for therapeutic management of various bacterial infections in humans and animals. The AMR genes can be exchanged through horizontal gene transfer within the Aeromonas spp. and between different genus of bacteria. Among the various aquatic and terrestrial animal species and human beings the infection may occur. |
|
8 |
Line 68-74. The easiness of incubation is repeated twice in two sentences. Could the authors provide more explanations for the application of Aeromonas spp. as AMR indicators in the environment? |
The World Health Organization (WHO) employs multidisciplinary strategies to explore the widespread issue of antimicrobial resistance (AMR). For this purpose a universal and manageable indicator bacteria species is needed. The genus Aeromonas is considered ideal to study antibiotic resistance gene (ARG) transmission dynamics due to its presence in various interconnected ecosystems. Members of the Aeromonas genus infect a broad range of hosts, including both cold- and warm-blooded animals, including humans. |
|
9 |
Line 74. How prevalent is Aeromonas spp. in terrestrial ecosystems? |
The Aeromonas spp. are reported from both diseased and apparently healthy human and animal species (Zhou et al., 2019). Also the bacteria is isolated from animal products and vegetable source (Kon et al., 2023). |
|
10 |
Line 80. FAO. Please provide the full name for an abbreviation. |
Modified and included in the text as per suggestion |
|
11 |
Line 110. What do the authors aim to detect with "phenotypic screening of ESBL producing Aeromonas spp."? |
ESBLs producing ability of bacteria confer resistance to a broad range of antibiotics. Detection of ESBLs positive bacteria contributes to epidemiological research and monitoring its prevalence which is essential in understanding AMR patterns in microbial populations. |
|
12 |
Materials and Methods. Please add the period when samples were collected/the study was conducted. Add no. of samples per location? How many sampling sessions were in total? How many mussels were collected? From water or cost? |
Monthly samplings were performed from January, 2023 to June 2023 over a period of six months. Samplings were done from three waterbodies connected to the sewage canals. A total of 105 mussels were collected for the study. The above sentence was added in materials and methods as per the suggestion. |
|
13 |
2.2. In paper no. 17 biochemical confirmation procedure of Aeromonas spp. isolates is not described. |
Checked the cross reference. The paper cited is related to bacteria isolates of bivalve origin and the original biochemical tests are described in the cross reference (Abbot et al., 2003) cited by Dahanayake et al., 2019. To clarify the same we have described the tests in the materials and method section. |
|
14 |
Very confusing that the authors provide a reference to Table 1 only for 2.2. subsection. PCR mix and reaction conditions are not provided. |
Table 1 is also cited in the 2.6 subsection for the detection of ESBL genes. PCR mix and the reaction condition for amplification of Aeromonas genus specific 16s rRNA and ESBL gene are provided separately in the respective section.
|
|
15 |
2.3. Please provide culturing conditions for inoculated TSA and MHA agars. Please check manufacturer information for all microbiological media. |
Incubation temperature and period are provided in the section 2.3. Manufacturer information for all microbiological media have also been checked and found in right order. |
|
16 |
2.3., 2.5. Please merge phenotypical examinations of ESBL isolates in one subsection. |
Based on the Reviewer's suggestion the preliminary screening test for ESBL from Section 2.3 has been shifted to 2.5 Section and named as Phenotypic screening and confirmation of ESBL production. Now the 2.3 section contains only antimicrobial sensitivity testing. |
|
17 |
Results.
3.1. Biochemical procedures were not described in the M and M section. |
The biochemical tests were mentioned in the material and methods part as per the suggestion. Line no 155-158.
|
|
18 |
3.2. Difficult to follow results on antimicrobial resistance. Please show them in a table. |
The stack chart is converted to a table representing the antibiotic susceptibility pattern of Aeromonas spp. |
|
19 |
Regarding 92 isolates, how were they obtained? From culturing or they were PCR confirmed? How many isolates were confirmed from one sample? |
92 isolates were 1st isolated by specific Aeromonas isolation broth and agar media then confirmed by genus specific 16s rRNA amplifications. From one sample one bacteria was taken for further confirmation and analysis. |
|
20 |
Line 229. P less than 0.05 please use symbols. |
As per the suggestion “P less than 0.05” are replaced by “P < 0.05” and corrected in all text. |
|
21 |
Fig.2. |
|
|
22 |
Fig.4. What is the difference between ESBL(-) and ESBL(+) and negative and positive? |
In place of ESBL(-) and ESBL(+), negative and positive was written by mistake. Corrected in the figure 4 as per the suggestion. |
|
23 |
Line 254. I cannot find bluish colonies in Fig.5. |
ESBL producing Aeromonas spp. were cultivated on ESBL isolation media (HiMedia, India) and give blue to purple colonies. Figure no was written 5 in place of 6 in the text after the modifications. |
|
24 |
Fig.6. any specific detail on antibiotics, incubation conditions, or isolates is not provided. |
Figure 6 is renamed figure 3 after the rearrangement and the stack chart is changed to a table. This figure shows the AST for all the mentioned antibiotics in the disc method, screening of ESBL producing Aeromonas spp. and confirmation by CDT method for representative samples. Incubation conditions are provided in the Material Method section. The same has also been mentioned in the figure caption as suggested. |
|
25 |
Table 2. Explanations for antimicrobials are not provided. What do the authors aim to show with this table? Information on specific ESBL determinants could be included. |
After modification, table 2 is now changed to table 3 and this table is explained in the result section 3.5. Where the resistance pattern of different combinations of antibiotics is given. Information on specific ESBL determinants against each isolate is now included in a separate column in the same table. |
|
26 |
Merge 3.6. and 3.7. subsections. |
3.6 and 3.7 subsections are merged and named as “ESBL genes and their co-occurrence harboured by phenotypically confirmed ESBL Aeromonas spp.” under 3.6. |
|
27 |
3.8. subsection should be merged with 3.5., 3.6. and 3.7. |
3.6 and 3.7 subsections are merged however 3.5 and 3.8 are kept separately as 3.5 explaining about the antibiotic resistance pattern to all antibiotics against ESBL producer Aeromonas spp. In the 3.8 subsection, which is modified to 3.7 after merging of 3.6 and 3.7 explain the ESBL positive Aeromonas spp. not only exhibit resistance to antibiotics within the cephalosporin group but also demonstrate resistance to antibiotics outside this specific class along with the co-occurrence of ESBL genes in these strains. |
|
28 |
Discussion.
Line 323. "these bacterial spp. poses". Please explain. |
Contamination with different spp of Aeromonas poses a risk to human health, especially through the consumption of aquaculture-derived food |
|
29 |
Line 330-331. Please add information on foodborne outbreaks of Aeromonas spp. caused by resistant Aeromonas spp. |
The related report is included in the discussion section with reference “Zhang et al. (2012) reported that antibiotic-resistant A. hydrophila in vegetables through contaminated water has been linked to significant outbreaks, with substantial health risk in China”. The reference is now cited as 34. |
|
30 |
Line 335-336. In mussels? In waters? Not clear how this relates to the present manuscript. |
We have worked on microbe isolates of mussels/bivalves origin from sewage fed wetland. Through discussion, we tried to correlate the organic load and contamination in such wetland and antimicrobial resistance patterns. |
|
31 |
Line 336-337. The present study aimed to detect ESBL Aeromonas spp. in Lamellides but not in the environment. |
The present study aimed to detect ESBL producing Aeromonas spp. in L. maginalis, as these bivalves are suspension feeders and concentrate contamination and surrounding bacteria present in the water bodies. |
|
32 |
Line 338-339. Have the authors studied ESBL production in Aeromonas spp. isolates? What kind of concerns? From which foods and food-production animals were resistant Aeromonas spp. isolated? Have the authors draw this conclusion from one study? |
The study was conducted on the ESBL production in Aeromonas spp., isolated from L. marginalis.
The concern highlighted in the statement is related to the presence of MDR Aeromonas strains in a contaminated environment. Additionally, ESBL-producing Aeromonas strains were identified, further intensifying the public health concern. The combination of MDR and ESBL resistance in Aeromonas strains suggests a heightened level of antibiotic resistance, posing a substantial global public health concern. The potential spread of such antibiotic-resistant bacteria could lead to increased morbidity and mortality, as well as greater difficulties in controlling infectious diseases.
There are reports on drug resistant Aeromonas spp. from different sources like humans, fish, vegetables, meat, milk, cheeses, chickens, and bivalves (Kon et a., 2023; Lee et al., 2021; Plassard et al., 2021; Wang et al., 2020; Hammad et al., 2018; Deng et al., 2014).
The conclusion was drawn based on these previous studies and our present findings |
|
33 |
Line 351. What kind of findings? |
The study by Sola et al., 2022, highlighted two main concerns. First, it addressed the issue of environmental contamination caused by various effluents, which could introduce antibiotic-resistant bacteria into the aquatic environment. Second, the study pointed out the potential risk of establishing a reservoir of antibiotic-resistant bacteria in bivalves and seafood products that are meant for human consumption. These findings underscore the importance of addressing environmental contamination and the potential impact on food safety and public health. |
|
34 |
Line 353. Non beta-lactam antibiotics...carbapenems. Please check. |
As carbapenem antibiotics come under â-lactam antibiotics, the text is changed to “Other than cephalosporins group of antibiotics”. |
|
35 |
Line 358-360. The sentence is unclear. |
Carbapenem antibiotic is the option for treating bacteria resistant to cephalosporins. In our study we found that ESBL producing Aeromonas spp were resistant to both the classes of antibiotics, indicating a great concern in treating diseases caused by these resistant strains. This is added in the discussion section at line no.456-459 |
|
36 |
Line 397-398. How this was addressed by the present study? |
In our current study, we noticed that Aeromonas bacteria carrying the CTX-M gene tended to be resistant to cefoxitin, a second-generation cephalosporin. But a prior study conducted by Singh et al. (2020), focusing on ESBL-producing E. coli, found that most of the bacteria in their study were susceptible to cefoxitin.
Porins, classified as outer membrane proteins (OMPs), play a vital role in the outer membranes of gram-negative bacteria. Resistance to β-lactam antibiotics in these bacteria may result from the loss of particular porins, leading to a reduction in the permeability of the outer membrane to β-lactam antibiotics. In Klebsiella pneumoniae, two significant non-specific porins, Omp K35 and Omp K36, have been identified. Omp K35 and Omp K36 are analogous to E. coli's porins OmpF and OmpC, respectively. |
|
37 |
The discussion describes general information about ESBL-producing microorganisms with little relevance to the present study. |
Discussion sections has been modified as suggested by the reviewer. |
|
38 |
Conclusions.
Line 404-408. Two sentences with similar meaning. |
The sentences were merged and rewritten as follows:
Our study highlights the considerable presence of ESBL-producing Aeromonas spp. within bivalve molluscs and potentials in disseminating antibiotic resistance. The findings on antimicrobial resistance in edible bivalve-associated Aeromonas spp., further warn on food safety and public health hazards. |
|
39 |
Aeromonas spp. are autochthonous microorganisms to aquatic environments. How was it highlighted in the Introduction, Discussion and Conclusions? |
As per suggestion of the reviewer the related statements are included in the Introduction and discussion: “Most of the bacteria species under the genus Aeromonas exist in the aquatic system without causing any harm. Some of the Aeromonas spp. are considered as an opportunistic pathogen and cause diseases in both aquatic and terrestrial animals when environmental and host condition favours”. (at line no. 62-66)
“Aeromonas spp. as natural contaminants in bivalves and their surroundings [29], contamination with these bacterial spp. poses a risk to human health, especially through the consumption of aquaculture-derived food [23]” ( at line no. 418-420) |

Reviewer 4 Report
Comments and Suggestions for Authors
The manuscriipt entitled "Prevalence of Extended-Spectrum β-Lactamases (ESBLs) Producing Aeromonas spp. Isolated From Lamellidens marginalis (Lamark, 1819) of Sewage Fed Wetland: A Phenotypic and Genotypic Approach" by Mohanty and co-authors describe the isolation of Aeromonas spp. from Lamellidens marginalis cultured in freshwaters wetlands fed with sewage, and their characterization concerning resistance to antibiotics and presence of selected ESBL- encoded genes. The authors concluded that L. marginalis could act as reservoirs of antibiotics resistance genes, representing a serious threat to human populations consuming this mussel bivalves. In brief the authors describe sampling, isolation of bacteria and identification as Aeromonas spp., characterization of antibiotics resistance profiles, screening for the presence of ESBL, and PCR to confirm the presence of genes encoding 3 selected ESBLs. The manuscript is well written and organized, and the study is on the scope of the journal. There are a few questions which the authors should address:
Why only 3 ESBL encoding genes were selected? Is there any indication that the other ESBL- encoded genes are less frequent in Aeromonas?
line 59: The authors state that "This bacteria has 58 gained recognition as a promising candidate due to its ability to thrive in both host and 59 their surrounding environment." the bacterium is a good candidate for what?
lines 154 and 158: beta lactamase inhibitors? while the first are beta-lactams, the others are mixtures of beta-lactams and beta-lactamase inhibitors.
lines 187-189: The reaction mixtures are incomplete. Instead, describe the composition of the reaction mixture.
Figure 4: Why panels A and B are labeled as ESBL(-) and ESBL(+) and the other panels are labeled as negative and positive
line 244: delete the "a" in the sentence "in a figure 5".
line 328: correct as follows: "... molluscs such as clams, mussels, and oysters..."
line 395: blaCTX-M resistant?
404: ESBL resistant?
Comments on the Quality of English LanguageThe English quality is fine, no need of corrections.
Author Response
|
The authors are highly thankful to the reviewer for relevant suggestions to improve the manuscript |
||
|
sl no |
comment |
Response |
|
1 |
Why only 3 ESBL encoding genes were selected? Is there any indication that the other ESBL- encoded genes are less frequent in Aeromonas? |
ESBL-encoding genes are diverse and can be categorized into Ambler class A ESBL (ESBLA), miscellaneous ESBL (ESBLM), and ESBLs that degrade carbapenems (ESBLCARBA). The ESBLA group, which can hydrolyze a wide range of beta-lactam antibiotics encompassing SHV, TEM, and CTX-M β-lactamases, is noted for their global prevalence and emerging significance. Since data on the prevalence of these ESBLA genes in Aeromonas spp. isolated from benthic organisms (freshwater mollusk) within the specific region are lacking, the study was initiated to address this gap. However, point raised by the reviewer under taken for future research work. |
|
2 |
line 59: The authors state that "This bacteria has gained recognition as a promising candidate due to its ability to thrive in both host and their surrounding environment." the bacterium is a good candidate for what? |
Aeromonas spp., widely distributed across various environments and serves as both a pathogenic entity and nonpathogenic bacteria. It serves as a reservoir for antibiotic resistance genes that makes them a good candidate for screening and characterizing antibiotic resistance profiles.
|
|
3 |
lines 154 and 158: beta lactamase inhibitors? while the first are beta-lactams, the others are mixtures of beta-lactams and beta-lactamase inhibitors. |
Modified in the text as per the suggestion. The first category are beta-lactam antibiotics, whereas second category contains a combination with beta-lactamase inhibitors as one of the component. Hence, the second category named as beta-lactamase inhibitor as per reference Lima et al., 2020. |
|
4 |
lines 187-189: The reaction mixtures are incomplete. Instead, describe the composition of the reaction mixture. |
Modified in the text as per the suggestion. |
|
5 |
Figure 4: Why panels A and B are labeled as ESBL(-) and ESBL(+) and the other panels are labeled as negative and positive |
In the figure 4, the corrections have been made as per suggestion. To maintain the uniformity, the labeling has been made ESBL(-) and ESBL(+) in all the panels. |
|
6 |
line 244: delete the "a" in the sentence "in a figure 5". |
Deleted in the text. And figure no is also modified after the figure 2 is changed to table. |
|
7 |
line 328: correct as follows: "... molluscs such as clams, mussels, and oysters..." |
Corrected in the text |
|
8 |
line 395: blaCTX-M resistant? |
The word “resistant” is replaced by “carrying” and the whole sentence is modified. |
|
9 |
404: ESBL resistant? |
ESBL resistant is replaced by ESBL producing |

Round 2
Reviewer 1 Report
Comments and Suggestions for Authors
The manuscript has improved considerably. However, some further polishing of the text is necessary. I have attached a file, where I have added sticky notes where changes need to be done.

I have attached a file, where I have highlighted with blue words and spots, where the grammar or use of a word needs to be reconsidered.
Author Response
|
Reviewer 1: The authors are highly obliged for your important corrections to improve the manuscript |
||
|
1 |
Our study highlights the considerable presence of ESBL-producing Aeromonas spp. within bivalve molluscs and potentials in disseminating antibiotic resistance. |
The sentence is changed to “Our study highlights the presence of ESBL-producing Aeromonas spp. within bivalve molluscs and their potential role in disseminating antibiotic resistance.” |
|
2 |
The present investigation on antimicrobial resistance of microbes from freshwater bivalve origin leads to the important findings for a particular geographical region. Hence, we opine that the patterns of resistance may differ between geographic regions and origin of the microbes. |
The sentence is changed and rewritten as follows:
“The present investigation on antimicrobial resistance of microbes from freshwater bivalve originresults in significant findings specific to a particular geographic area”. Hence, we believe that the patterns of resistance may differ between geographic regions and origin of the microbes. |
|
3 |
The observed resistance patterns within Aeromonas spp. provided hints towards contamination of L. marginalis from anthropogenic sources. |
The word “hints” is replaced with “evidence” |
|
4 |
Instead of giving the names of authors tell where the study was done |
All the cited studies were conducted in Korea. All the authors names were deleted and modified as per the suggestion. |
|
5 |
In our study we found that ESBL producing Aeromonas spp were resistant to both the classes of antibiotics, indicating a great concern in treating diseases caused by these resistant strains [47] |
The word “both the classes of antibiotics” is changed to “both classes of antibiotics” |
|
6 |
write consistently ESBL-producing |
The word “ESBL producing” are replaced by “ESBL-producing” |
|
7 |
delete the names in the text |
The author name is deleted as per the suggestion. |
|
8 |
fig |
“fig” replaced by Fig |
|
9 |
give the number of tested isolates |
The total no of 92 isolates mentioned in the tittle if the table 2 |
|
10 |
open the abbreviations up in the footnote of the table. |
The footnote is added in the table 2 as per the suggestion. |
|
11 |
All the results are given in Table2. |
The sentence is modified as per the suggestion. |
|
12 |
delete all the data that is anderlined, it is now given in the table. |
Deleted all the data as per suggestion. |
|
13 |
Again, give concentrations, not just volumes. |
The concentration are added in the text |
|
14 |
three set |
The word three set is replaced by “three sets” |
|
15 |
presumptive ESBL-positive. For further confirmation the |
The word seems to be grammatically correct. |
|
16 |
give concentrations rather than volumes |
The volumes are deleted from the text and the concentration of the same is added. |
|
11 |
Sampling were done from three waterbodies connected to the sewage canals. |
“Samplings were done” |
|
10 |
aimed to isolate the bacteria from edible bivalves from sewage fed wetland and point out a different source of antibiotic resistant microbes in the aquatic system with great potentials for fisheries resources. |
The word “the” is deleted and the sentence is rewritten as aimed to isolate bacteria from edible bivalves from sewage fed wetland and identify antibiotic resistant microbes from different source of the aquatic ecosystem with great potentials for fisheries resources. |
|
9 |
Bivalve molluscs have the potential to be contaminated with these bacterial pathogens, posing significant safety risks when consumed raw or undercooked, as it can lead to human illness. |
The word “with” is replaced by “by” |
|
8 |
Food and Aggriculture Organisation |
The word is replaced as “Food and Agriculture Organisation” |
|
7 |
This bacterium has become a promising candidate due to its ability to grow in bothhost, surrounding environment and artificial media in laboratory condition |
The word both is deleted and sentence is modified as “This bacterium has become a promising candidate due to its ability to grow in host, surrounding environment and artificial media in laboratory conditions” |
|
5 |
Most of the bacteria species under the genus Aeromonas exist in the aquatic system without causing any harm. |
|
|
6 |
Some of the Aeromonas spp. are considered as opportunistic pathogen and cause diseases in both aquatic and terrestrial animals when environmental and host condition favours. |
The sentence is changed to “Some of the Aeromonas spp. are considered as opportunistic pathogen and cause diseases in both aquatic and terrestrial animals when environmental and host conditions are favourable” |
|
4 |
Aeromonas spp. exist in wide host range with high significance in AMR and noticeable resistance against polymixin B - a highest priority critically important antimicrobials for human medicine” |
The sentence is changed and rewrite as: Aeromonas spp. exist in wide host range and hold considerable importance in the context of AMR and and noticeable resistance against polymixin B - a highest priority critically important antimicrobials for human medicine” |
|
1 |
In the present investigation 92 isolates of Aeromonas spp. were recovered from 105 bivalvessamples and screened for their antimicrobial resistance patterns. |
“Samples” word is deleted and rewrite as “105 bivalves and screened for their antimicrobial resistance patterns” |
|
2 |
In vitro antibiotic resistance profiling showed a higher Multiple Antibiotic resistance (MAR) index of 0.8 with the highest resistance against ampicillin/sulbactam |
The word “resistance” is replaced with “Resistance” |
|
3 |
The identification of extended-spectrum-β-lactamases (ESBLs) genes further demands the necessity of continuous surveillance and systematic monitoring, considering its perspectives of potential health risks for both animals and human beings. |
The word further and perspectives are deleted from the text. |

Reviewer 2 Report
Comments and Suggestions for Authors
The authors revised their manuscript on the prevalence of extended-spectrum β-lactamases (ESBLs) producing Aeromonas spp. isolated from Lamellidens marginalis (Lamark, 1819) of sewage fed wetland. However, this revision avoided most of the severe concerns and focused only on minor items. There is still no justification why this work could increase our understanding of antibiotic resistance. This justification has to be made in the context of a multitude of similar reports on the antibiotic resistances of Aeromonas spp. and the responsible genes. Here are some of them and none of them has been addressed in the manuscript:
Ko, W. C., Yu, K. W., Liu, C. Y., Huang, C. T., Leu, H. S., & Chuang, Y. C. (1996). Increasing antibiotic resistance in clinical isolates of Aeromonas strains in Taiwan. Antimicrobial agents and chemotherapy, 40(5), 1260-1262.
Piotrowska, M., & Popowska, M. (2014). The prevalence of antibiotic resistance genes among Aeromonas species in aquatic environments. Annals of microbiology, 64, 921-934.
Yucel, N., Aslim, B. E. L. M. A., & Beyatli, Y. (2005). Prevalence and resistance to antibiotics for Aeromonas species isolated from retail fish in Turkey. Journal of food quality, 28(4), 313-324.
Piotrowska, M., & Popowska, M. (2015). Insight into the mobilome of Aeromonas strains. Frontiers in microbiology, 6, 494.
Deng, Y. T., Wu, Y. L., Tan, A. P., Huang, Y. P., Jiang, L., Xue, H. J., ... & Zhao, F. (2014). Analysis of antimicrobial resistance genes in Aeromonas spp. isolated from cultured freshwater animals in China. Microbial Drug Resistance, 20(4), 350-356.
Shuang, M. E. N. G., Wang, Y. L., LIU, C. G., Jing, Y. A. N. G., Min, Y. U. A. N., Bai, X. N., ... & Juan, L. I. (2020). Genetic diversity, antimicrobial resistance, and virulence genes of Aeromonas isolates from clinical patients, tap water systems, and food. Biomedical and Environmental Sciences, 33(6), 385-395.
Fauzi, N. N. F. N. M., Hamdan, R. H., Mohamed, M., Ismail, A., Zin, A. A. M., & Mohamad, N. F. A. (2021). Prevalence, antibiotic susceptibility, and presence of drug resistance genes in Aeromonas spp. isolated from freshwater fish in Kelantan and Terengganu states, Malaysia. Veterinary World, 14(8), 2064.
Hayatgheib, N., Calvez, S., Fournel, C., Pineau, L., Pouliquen, H., & Moreau, E. (2021). Antimicrobial susceptibility profiles and resistance genes in genus Aeromonas spp. isolated from the environment and rainbow trout of two fish farms in France. Microorganisms, 9(6), 1201.
Dahanayake, P. S., Hossain, S., Wickramanayake, M. V. K. S., & Heo, G. J. (2020). Prevalence of virulence and antimicrobial resistance genes in Aeromonas species isolated from marketed cockles (Tegillarca granosa) in Korea. Letters in Applied Microbiology, 71(1), 94-101.
Results based on only 55 isolates are not impressive and some of these results are probably not significant. These results have to be discussed in the context of results from other reports (such as those above). What is different, what is novel?
Figure 4J: is the difference really significant?
Because of the small number of samples, decimals in the percentages do not make sense. This has still not been corrected everywhere, e.g. line 508.
Comments on the Quality of English Languagesome typos and grammar mistakes
Author Response
|
Reviewer 2: We are thankful for your suggestions. Now we have improved the manuscript, considering the concerns raised by you. |
||
|
The authors revised their manuscript on the prevalence of extended-spectrum β-lactamases (ESBLs) producing Aeromonas spp. isolated from Lamellidens marginalis (Lamark, 1819) of sewage fed wetland. However, this revision avoided most of the severe concerns and focused only on minor items. There is still no justification why this work could increase our understanding of antibiotic resistance. This justification has to be made in the context of a multitude of similar reports on the antibiotic resistances of Aeromonas spp. and the responsible genes. Here are some of them and none of them has been addressed in the manuscript: |
The references cited by the Reviewer are having link with our work, but some of the literatures are review article [Piotrowska et al (2014 and2015)], some studies are conducted on the bacteria isolated from human as clinical isolates [Ko et al (1996), Shuang et al. (2020)], market fish (4), fish and aquatic system [Hayatgheib et al (2021), Xue et al. (2014), Fauzi et al (2021)] and cockles (Dahanayake et al. (2020)] from different geographical region including Australia, Turkey, China and Korea. Our approach was to isolate the bacteria from sewage fed fresh water origin bivalve. From this point of view our study differs from all these studies. Only one study conducted by (9) on cockles, ie a marine bivalve. Moreover, this emerging field of research need continuous update on epidemiological reports on drug resistance in various bacteria across the terrestrial and aquatic animal and environmental origin. Hence, our finding will add some knowledge of drug resistance patter in Aeromonas isolates from freshwater edible bivalve from this agroclimatic domain of India. Spatio-temporal monitoring of drug resistance and cumulative data on different animal origin will be helpful for better understanding and future planning on mitigation of this burning problem. In our opinion more such reports are required from different geographical regions for better conclusions. |
|
|
Results based on only 55 isolates are not impressive and some of these results are probably not significant. These results have to be discussed in the context of results from other reports (such as those above). What is different, what is novel? |
We are agreed that larger sample size is always expected to draw clear conclusions. In this study, a preliminary screening was done for a specific type of aquatic system as described earlier. Samples were taken from 105 bivalves. A total of 92 nos Aeromonas spp. were isolated from 105 bivalves. We screened 92 bacteria isolates. Out of which 55 isolates were phenotypically ESBL-producers. Then from 55 isolates, 51 isolates were confirmed as ESBL-producer by Combine disc Test (CDT) method. Hence, the findings out of 55 isolates are sufficient to represent an aquatic system.
Results from the other reports like Ko et al., 1996; Shuang et al., 2020; Deng et al., 2014; Yucel et al., 2005; Dhannayake et al., 2020 in addition to previous references are compared for antibiotic resistance, MAR index and ESBL gene prevalence. Those were suitably discussed and incorporated in the related sections as per suggestion. Please refer to line no 474-475; 491-492;509-510; 515-524; 534-540; 549-554. |
|
|
Figure 4: is the difference really significant? |
The significance level is mentioned as * .The *** in the figure 4 refers to P<0.001, ** refers to P<0.01 and * refers to P<0.05. |
|
|
Because of the small number of samples, decimals in the percentages do not make sense. This has still not been corrected everywhere, e.g. line 508. |
All decimals numbers are rewritten as the nearest whole number |
|

Reviewer 4 Report
Comments and Suggestions for Authors
The revised version of the manuscript addressed most of the criticisms raised. There is however, a correction that should be done: the authors should indicate the concentration of DNA template and not the volume when describing the composition of PCR mixtures.
Author Response
|
Reviewer 4: The authors are highly thankful to the reviewer for relevant suggestions to improve the manuscript |
||
|
The authors should indicate the concentration of DNA template and not the volume when describing the composition of PCR mixtures. |
The concentrations are added in the text as per the suggestion. |
|
